# Skull-Base Surgery—A Narrative Review on Current Approaches and Future Developments in Surgical Navigation

**DOI:** 10.3390/jcm12072706

**Published:** 2023-04-04

**Authors:** Sharon Tzelnick, Vittorio Rampinelli, Axel Sahovaler, Leonardo Franz, Harley H. L. Chan, Michael J. Daly, Jonathan C. Irish

**Affiliations:** 1Division of Head and Neck Surgery, Princess Margaret Cancer Center, University of Toronto, Toronto, ON M5G 2M9, Canada; 2Guided Therapeutics (GTx) Program, TECHNA Institute, University Health Network, Toronto, ON M5G 2C4, Canada; 3Unit of Otorhinolaryngology—Head and Neck Surgery, Department of Medical and Surgical Specialties, Radiologic Sciences and Public Health, University of Brescia, 25121 Brescia, Italy; 4Technology for Health (PhD Program), Department of Information Engineering, University of Brescia, 25121 Brescia, Italy; 5Head & Neck Surgery Unit, University College London Hospitals, London NW1 2PG, UK; 6Department of Neuroscience DNS, Otolaryngology Section, University of Padova, 35122 Padua, Italy

**Keywords:** surgical navigation, skull-base surgery, augmented reality

## Abstract

Surgical navigation technology combines patient imaging studies with intraoperative real-time data to improve surgical precision and patient outcomes. The navigation workflow can also include preoperative planning, which can reliably simulate the intended resection and reconstruction. The advantage of this approach in skull-base surgery is that it guides access into a complex three-dimensional area and orients tumors intraoperatively with regard to critical structures, such as the orbit, carotid artery and brain. This enhances a surgeon’s capabilities to preserve normal anatomy while resecting tumors with adequate margins. The aim of this narrative review is to outline the state of the art and the future directions of surgical navigation in the skull base, focusing on the advantages and pitfalls of this technique. We will also present our group experience in this field, within the frame of the current research trends.

## 1. Introduction

Surgical navigation merges patient imaging with navigation software aiming to improve surgical precision. Traditionally, the workflow of this approach consisted of intraoperative guidance correlating patient images with the spatial anatomy through the use of a “navigated pointer” orienting the surgeon in areas with complex anatomy. Most recently, this technique incorporated a preoperative planning phase, which simulates surgical resection and/or reconstruction [1,2], and an intraoperative phase based on real-time navigation [3]. To accomplish this, volumetric images can be used to obtain a three-dimensional anatomical rendering that may be modified to simulate surgery or used as a virtual volume to be matched with the actual space of the surgical field during navigation.

Skull-base surgery is a very adequate field to implement surgical navigation, with challenges such as: (1) complex anatomy and critical structures in close proximity within a limited working space; (2) prolonged learning curve in endoscopic assisted procedures, with increased difficulty with real-life conditions such as bleeding, tumors and scar tissue obscuring anatomic landmarks; and (3) traditional employed two-dimensional tri-planar radiological views, limiting the surgeon’s ability to accurately gauge depth during the surgery [4]. These aspects are potential explanations for the complication rates of skull-base surgery reported in the literature [4,5,6,7]. To address these concerns, navigational systems have been developed to provide real-time tracking that can give accurate feedback to the surgeon, enhancing spatial awareness while reducing task workload during skull-base surgery [8,9,10].

Surgical navigation, together with improved surgical instrumentation and techniques, have helped in decreasing complication rates in skull-base functional surgery [11,12,13], and extended surgical indications in oncologic procedures [14,15,16,17,18,19,20,21,22]. Preclinical studies have been continuing to exploit new navigation technologies to guide tumor ablation and to anatomically orient the surgeon in three-dimensional planes. The aim of this narrative review is to outline the state of the art and the future directions of surgical navigation specifically in skull-base surgery, focusing on the advantages and pitfalls of these techniques. We will also present our groups’ experience in this field to demonstrate current research trends.

## 2. Surgical Navigation

### 2.1. General Concepts

The key concept of image-guided surgery/surgical navigation resides in the use of volumetric imaging data to create a road map for surgeons via enhanced intraoperative visualization of surgical sites and anatomical landmarks.

All image-guided systems share similar components: a tracking system, which detects the position of surgical instruments in the actual operative field, and a computer workstation, which matches the spatial coordinates of the instruments, as detected by the tracking system. Devices for tracking surgical tools include stereoscopic near-infrared cameras, electromagnetic sensors, video-based tracking and radio-frequency tags [23]. The position of the surgical instruments with reference to the preoperative images is displayed on a monitor during surgery.

Currently, there are two types of tracking systems based on either electromagnetic or optical technology [24] in the head and neck region. Electromagnetic tracking systems includes an emitter, which provides a magnetic field around the patients’ head, an electromagnetic reference, which is fixed in a steady position on the patients’ head, serving as an origin for the volumetric coordinates of the operative field, and a probe, whose position is tracked with regards to the reference [25]. Optical tracking is based on an infrared stereo-camera, which detects the position of a probe with regards to an optical reference, marked with reflective spheres or light-emitting diodes [26]. It is still debated which system provides better performances in terms of accuracy and ergonomics; however, both approaches may show an error range within a few millimeters [3,27,28,29,30]. In terms of potential pitfalls, interruptions in the line of sight between the stereo-camera and the reference and/or probe may result in a disturbance in the anatomical location in optical systems [25]. On the other hand, the intraoperative accuracy of electromagnetic systems may be affected by interferences due to the presence of large metal instruments. Regardless of the tracking technology utilized, the placement of an intraoperative reference localization device in a fixed position on the patients’ anatomy is a critical step. Once the reference point is fixed, the dynamic registration process allows for patient repositioning while maintaining a match of the virtual volume of the radiological images with the real three-dimensional space of the actual operative field.

### 2.2. Clinical Applications

Over the past 20 years, endoscopic skull-base surgeons have used surgical navigation technology both for functional and oncologic purposes. The skull base is exceedingly complex from an anatomic standpoint, and an orientation error of millimeters can lead to catastrophic bleeding and CSF leak. Currently, the latest navigation systems have shown an accuracy of <1.0 mm combined with intuitive and fast pre-use calibration procedures [31], representing an evolution in skull-base navigation technology.

Multiple studies have previously shown that image-guided surgery (IGS) lowers revision surgery and recurrence rates [32,33]. In a meta-analysis comparing endoscopic sinus surgery (ESS) with and without image-guided surgery by Vreugdenburg et al. [6], IGS use was associated with a decreased risk of major (odds ratio (OR) = 0.36; 95% CI 0.18–0.75), orbital (OR = 0.38; 95% CI 0.17–0.83), and total complications (OR = 0.58; 95% CI 0.37–0.92). In 2012, the American Academy of Otolaryngology—Head and Neck Surgery endorsed the use of IGS during ESS in select cases based on expert consensus opinion and literature evidence [34]. This includes the following: 1. Revision sinus surgery; 2. Distorted sinus anatomy of development, postoperative or traumatic origin; 3. Extensive sinonasal polyposis; 4. Pathology involving the frontal, posterior ethmoid and sphenoid sinuses; 5. Disease abutting the skull base, orbit, optic nerve or carotid artery; 6. CSF rhinorrhea or conditions in which there is a skull base defect; 7. Benign and malignant sinonasal neoplasms.

The advantages of navigation for pure skull-base procedures have been mainly reported for pituitary surgery [35,36,37]. In trans-sphenoidal approaches without well pneumatized sinuses or revisions, navigation was particularly useful in confirming the position of the internal carotid artery or in locating major neural structures, such as the optic nerve [35,36,37]. Despite moderate use of navigation technology in lateral skull-base surgery [30,38,39], reports have demonstrated operative time reduction, optimized exposure for surgical corridors, and increased safety in difficult cases [39]. Other skull-base surgery series, albeit with small samples, showed the advantage of the navigation system in the perioperative outcomes [11,40,41,42,43,44].

Navigation systems have proved to be useful for other head and neck procedures, such as maxillofacial trauma [45,46,47,48], orthognathics [49,50,51], surgical oncology [52,53,54], temporomandibular joint interventions [55,56], and midface reconstruction [57,58,59]. Preliminary studies with a limited number of patients suggest that surgical navigation may improve margin status in head and neck cancer. Catanzaro et al. [60] and Tarsitano et al. [61] demonstrated that intraoperative navigation improves tumor-free margin status in terms of deep margin status when added to the standard procedure for advanced maxillary, oral or orbital cancers. Ricotta et al. [62] assessed the improvement in surgical margins using a navigation-guided, volumetric resection method, in patients with advanced-stage maxillary tumors, and showed an overall lower positive margin rate in the navigated group compared to patients that were operated without the use of virtual surgical software and 3D tumor segmentation.

### 2.3. Intraoperative Imaging

Surgical navigation most commonly relies on pre-operative imaging, and this can represent an inherent limitation of the traditional approaches, as changes in anatomy can occur since the preoperative images were obtained. A study by Strauss et al. [63] examined the compliance to image-guided surgery technology, and showed that 50% of surgeons changed their pre-planned surgical strategy intraoperatively during functional endoscopic sinus surgeries. Furthermore, other experiences that evaluated the advantage of intraoperative CT imaging in ESS showed that in approximately 25% of cases, the use of intraoperative CT modified the decision making and led to additional interventions [64,65]. A study by Muhanna et al. [66] aimed to assess the image quality of sinus and skull-base anatomical landmarks in surgical navigation using the intraoperative cone-beam CT to assist skull-base surgery. They showed a high bony detail image quality of intraoperative CBCT scanning in advanced skull-base surgery with improve visualization of vasculature using intravenous contrast.

Using intraoperative magnetic resonance imaging (iMRI) has been utilized to ensure maximal resection in pituitary adenomas, and demonstrated an improvement in the rate of gross total removal, detecting tumor remnants, and increasing progression-free survival [67,68,69]. Studies that evaluated iMRI in skull-base surgeries [70,71] also showed a higher gross total resection using the iMRI. A study by Ashour et al. [70] retrospectively reviewed 23 patients that underwent skull-base surgery with iMRI for a variety of pathologies (meningiomas, pituitary adenomas and acoustic neuromas) and showed a 25% additional tumor resection rate using iMRI. Metwali et al. [71] showed additional tumor resection rate of up to 50% in patients with skull-base chordomas guided by iMRI.

### 2.4. Surgical Navigation and Augmented Reality

A significant disadvantage of surgical navigation approaches is that the surgeon needs to blend the information displayed in monitors into the surgical field. Augmented reality (AR) is the fusion of virtual information in the real environment, and complements and integrates the concepts of traditional surgical navigation, providing a real-time anatomically detailed 3D virtual model, based on preoperative imaging data, overlaid on the real surgical field [72]. AR can expand the limited visual field of the nasal and skull-base endoscopic view, allowing surgeons to view deep anatomical structures, such as tumors, blood vessels, the brain, and orbits, in their original forms on top of the superficial surgical field. These real-time images can be projected either into a headset [73] or the endoscope screen [74]. Studies that examined the AR system in skull-base surgery have showed a high-accuracy 3D image-based registration [75,76,77]. A study by Lai et al. [74] examined an AR surgical navigation system (ARSN) with 3D intra-operative CBCT images that were fused with the view of the surgical field obtained by the endoscope camera. Their accuracy of the overlay, measured as mean target registration error (TRE), was 0.55 mm with a standard deviation of 0.24 mm. Li et al. [76] also showed a lower operative time using the AR system due to improved display, which facilitates the cognitive processes required to connect imaging data to real structures and eliminates the need to look away from the screen or use probes to verify surgical sites.

A study by Zeiger et al. [78] presented the first clinical implementation of a novel augmented reality endoscopic system coined EndoSNAP (Surgical Theater, Mayfield Village, Ohio). This visualization system allows the endoscopic surgeon to create and enhance 3D digital reconstructions before surgery based on radiology scans. Then, during endoscopic endonasal surgery, EndoSNAP links to the IGS system. The endoscope itself is tracked and produces an image of the 3D reconstruction that matches the video feed from the endoscope and can be projected adjacent to it. A total of 134 anterior skull-base cases ranging from pituitary adenomas to sinonasal disease and cerebrospinal fluid leaks were included using this novel mixed reality platform. Although, in this study, the accuracy of the system was not captured quantitatively, surgeons subjectively reported that the EndoSNAP system visualization helped them comprehend the relationships between vital structures, which helped them to be more time-efficient regarding the proximity to critical structures, such as the carotid.

The head mounted display (HMD) technology leverage AR into open surgery. This can be used either as video see-through (VST) HMDs, through a wearable display, or optical see-through (OST) HMDs with a direct view of the real world that optically merged with the virtual content [79]. Cercenelli et al. [80] presented an early prototype of Video and Optical See-Through Augmented Reality Surgical System (VOSTARS) AR wearable for open maxillofacial surgeries with a submillimetric accuracy in tracing osteotomy trajectories.

## 3. The GTx Lab Experience

Our group has been working with an in-house navigation software package (GTx-Eyes) [81] mostly in pre-clinical settings [10,82,83,84,85,86,87,88,89,90,91,92], and in proof-of-principle clinical studies [66,86,93,94,95,96] with proven benefits.

Two main lines of research have been conducted: one focused on quantifying the working “volume” of surgical approaches, the other aimed at predicting cutting trajectories providing adequate margins and developing cutting guides.

The GTx-Eyes software displays a three-dimensional image of a cone-beam computed tomography (CBCT) obtained from cadaver specimens or artificial skull models [88,97]. This technology allows the surgeon to locate a registered instrument or pointer tool superimposed on two-dimensional tri-planar views (e.g., axial, sagittal, coronal).

### 3.1. Objective Quazntification of Surgical Volumes

A surgical approach can be conceived as a truncated pyramid, with a superficial surface (access area) representing the area through which instruments are introduced in the patient, and a deep surface, the area exposed by the approach [88]. For example, during a transsphenoidal endoscopic approach in a cadaver, the superficial surface is represented by the nostril at the level of the pyriform aperture, while the deep surface corresponds to the posterior wall of the sphenoid sinus exposed and reached by the instrumentation [89]. ApproachViewer, part of the GTx software package, allows for the real-time registration of deep and superficial surfaces using a pointer to track their perimeters, thus providing visualization and quantification of the surgical pyramid in axial, coronal, and sagittal sections, as a 3D rendering while performing the cadaver dissection (Figure 1) [89].

Furthermore, in the post-dissection phase, it is possible to draw areas of interest on CT scans, with a sequential contouring. ApproachViewer matches each surgical pyramid with each area of interest, providing the absolute and percentage values of target areas exposed [88,89]. The volumetric and target areas analysis allows for an objective comparison of surgical approaches. With this technology, it is possible to compare approaches that exploit different corridors (i.e., endoscopic vs. open). Furthermore, it is possible to develop and classify approaches based on the increasing grades of surgical invasiveness.

Belotti et al., for instance, quantitatively compared four endoscopic endonasal trans-sphenoidal approaches to the sella and parasellar regions (hemi-sphenoidotomy, trans-rostral, extended trans-rostral with superior turbinectomy, and extended trans-rostral approach with posterior ethmoidectomy) [89]. The main findings were that hemi-sphenoidotomy provided limited exposure of the sellar area and a small working volume. The trans-rostral approach exposed the entire sellar area, while for exposure of lateral parasellar areas, superior turbinectomy and/or posterior ethmoidectomy were required [89].

Rampinelli et al. measured working volumes and exposure of key areas of the middle cranial fossa provided by the endoscope-assisted subtemporal key-hole epidural approach (ESKEA). A quantification of the working volume and exposure of four regions (sphenoorbital, parasellar, superior petrous apex, and squamopetrous) was performed, testing three incremental degrees of temporal dural retraction. Three modular corridors with incremental surgical invasiveness have been developed and described, with specific working volumes influenced by the degree of temporal lobe retraction, and exposure of different middle cranial fossa areas [90].

### 3.2. Development of Cutting Guidance System

One of the key novelties of the GTx-Eyes system is that it also introduces planar cutting tool capabilities along with three-dimensional (3D) volume rendering, allowing for visualization of the entire trajectory of the cutting instrument with respect to the tumor in 3D views (Figure 2). The tracking is provided by a stereoscopic infrared camera (Polaris Spectra, NDI, Waterloo, Ontario). Image-to-tracker registration is obtained by paired-point matching of pre-drilled divots by means of a tracked pointer, or alternatively can be achieved through an automatic registration technique [93]. In lab studies, a small four-sphere reference tool (NDI, Waterloo, Ontario) is anchored to the skull throughout registration and simulations with a registration error of 1 mm or less that is considered acceptable for all our navigation experiments.

Ferrari et al. [98] presented the benefits of three-dimensional (3D) navigation guidance for margin delineation during ablative open surgery for advanced sinonasal cancers with skull-base involvement. Using artificial skull tumor models in a simulation setting, the authors have shown that GTx-Eyes guidance significantly decreased intratumoral cut rates from 18.1% to 0.0% and improved margin delineation by 19.6% comparing unguided versus navigated simulations. Taboni et al. [99] further examined the same real-time tool tracking navigation system combined with 3D virtual endoscopy for the posterior maxillary sinus margin delineation. Much of the complexity of maxillary sinus surgery ablation is to ensure that the posterior osteotomy is posterior to the tumor margin. The complexity of determining the posterior osteotomy location is further complicated by the proximity to critical anatomical structures such as the internal carotid artery and neural structures within the pterygopalatine fossa. Therefore, the rationale of using 3D navigation is to provide real-time visualization of the tumor and these critical structures and to facilitate accurate positioning of the margin. In the study by Taboni et al. [99], a 2 mm alert cloud surrounding the carotid was added to the tumor-guided setting as a carotid-guided simulation. This was performed using an alarm reproducing the arterial flow sound when the trajectory of the navigated cutting tool, defining the posterior margin definition, was intersecting the proximity alert zone [8]. This study has shown that in 612 posterior margin transnasal delineations, 3D navigation decreased the frequency of positive posterior margins from 27% to 3%. Furthermore, based on our model, carotid injury was decreased from 41% to 15%. With these results, the added value of 3D rendering of the critical structures on virtual views and cross-sectional imaging with associated sound alerts was to increase the confidence of the surgeon during the procedure and help avoid simulated life-threatening complications.

### 3.3. Pre-Operative Virtual Planning

Along with the implementation of intraoperative image-guided surgery systems, virtual surgical planning protocols have been developing [28,29,100] to allow surgeons to obtain preoperative virtual resection planning. These can be reproduced intraoperatively with navigation assistance. This represents an innovative addition to the ability of tracking the entire trajectory of a cutting instrument, anticipating the margins that should be obtained postoperatively.

Preoperative virtual planning in head and neck cancer has been described over a decade ago: computer-aided design (CAD)/computer-aided manufacturing (CAM) led to the development of patient-specific, prefabricated cutting guides and reconstruction plate templates [101,102,103,104]. As a complement to these physical, in-hand cutting guides, our group has developed virtual cutting guides to our three-dimensional (3D) optical navigation. The advantages of virtual cutting guides over in-hand cutting guides include the ability to change the operative plan if tumor enlargement is identified with updated imaging before the surgery. In addition, this approach is possible where intraoperative physical guides cannot be adapted due to tumor extension.

### 3.4. Augmented Reality

Sahovaler et al. [105] reported a novel AR system in open sinonasal tumor resections in preclinical skull models and compared it to the more traditional intraoperative navigation systems. Using GTx-Eyes [81], tumors were projected by an external projector onto the skull surface. Optical sensors mounted to the projector case facilitated real-time tracking of the AR device to allow the projector and/or skull to be repositioned during tasks without compromising projection accuracy (Figure 3) [106,107].

AR technology showed improved margin delineation compared to unguided procedures (20.7% vs. 9.4%, *p* < 0.001). Furthermore, the main advantage of AR was that there was no need for increased screen time for the surgeons and they could concentrate purely on the surgical field. The study reported several innovations. First, tracking the AR projector as well as the projection surface with reflecting markers allowed for skull model and projector repositioning without losing accuracy. This is paramount in computer-assisted surgery, as it allows for precise projection even when movement occurs, as in real-time situations in the operating room. Second, the use of an external projector avoids the need for heavy wearable headsets by the surgeon.

A further modification to the AR system was performed [108] in order to improve intratumoral cut rates and overcome the challenges to obtain the correct angle between the projector and the projecting surface. Preoperative-planned maxillectomy osteotomies were designed using the surgical navigation platform (GTx-Eyes), and intraoperatively projected on the surgical field using the external projector AR in order to guide the surgeons. Furthermore, additional numerical cutting parameters, specifically distance to the osteotomy line and pitch and roll angles of the osteotome were projected onto the surgical field along with the 3D reconstructions of the tumors to indicate to the surgeon the virtual osteotomy direction with respect to the pre-determined resection plan. Information on these parameters was provided with a color-coded scale, green indicating adequate direction as continuous feedback for the surgeon throughout the osteotomies (Figure 4). With this further adjustment, the AR system showed significantly lower positive and close margins compared with the unguided simulations (0.0% vs. 1.9%, *p* < 0.0001; and 0.8% vs. 7.9%, *p* < 0.0001 respectively). Comparison between “ideal” pre-planned and AR osteotomies showed no difference. These data show that the AR approach enables guidance for all osteotomies, regardless of anatomical location, through the use of projected-navigation guides. Furthermore, since osteotomy lines are projected, soft tissues do not represent a limiting factor as in cases of placing 3D-printed guides on bone. Pre-operative planning can also be used to predict the postoperative defect more accurately as well.

Our experience with virtually planned osteotomies has also been extrapolated to other head and neck sites. Bernstein et al. [109] assessed the accuracy and reproducibility of 3D virtually planned osteotomies in mandible and maxilla models using GTx-Eyes with a navigated saw. The authors used surface rendering of the 3D-reconstructed CT scan and surface clipping (virtual removal) of the bone to one side of the saw plane, allowing the surgeon to see through the image of the bone, align the plane of the saw blade with the cut plan and judge the plane of the cut in distance, pitch, and roll in order to obtain a negative margin from the closest edge of the tumor within the bone (Figure 5).

Using data from a total of 448 osteotomies that were made by four surgeons across 12 mandibles and four skulls, this study shows that optical 3D-navigation had a median difference between the cut plan and all 3D-navigated osteotomies of 1.2 mm. More recently, a cadaveric study and a pilot clinical patient study of mandibulectomies and maxillectomies were performed in order to quantify the intra-operative navigation accuracy and to evaluate this technique under clinical conditions [98]. In five cadavers and five patients, a <1.5-mm accuracy between the planned cuts and the actual bone resection in post-resection imaging was seen.

Our navigation technology using an intraoperative, on-the-table, cone-beam computed tomography (CBCT) has been utilized in other clinical settings. Sahovaler et al. [93] published a pilot clinical study, including benign tumors in the femur, tibia and humerus and showed a mean target registration error of 0.83 ± 0.51 mm. We aim to replicate this approach in skull-base resections in the near future.

## 4. Conclusions and Future Directions

While the quality of evidence about the employment of navigation approaches compared to the traditional techniques is limited and lacking large-scale controlled trials, the available literature suggests that improved intraoperative accuracy may also result in a clinical benefit in terms of outcome and reduced complication risk.

Future studies, preferably in multi-center settings, should focus more on the outcome of patients treated with computer-aided surgery approaches, to confirm the actual contribution of these techniques on the overall outcome of the patients.

Moreover, the combined use of intraoperative CBCT imaging [74] and image-guided surgery techniques is likely to further increase the accuracy of skull-base surgery procedures, and we anticipate that this will be an emerging research trend over the coming years.

## Figures and Tables

**Figure 1 jcm-12-02706-f001:**
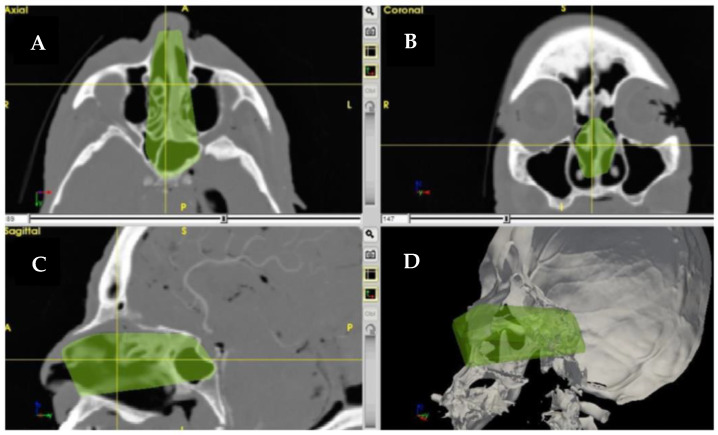
Rendering of the surgical pyramid in (**A**) axial, (**B**) coronal, (**C**) sagittal sections, and (**D**) 3D rendering for the endoscopic trans-rostral trans-sphenoidal approach.

**Figure 2 jcm-12-02706-f002:**
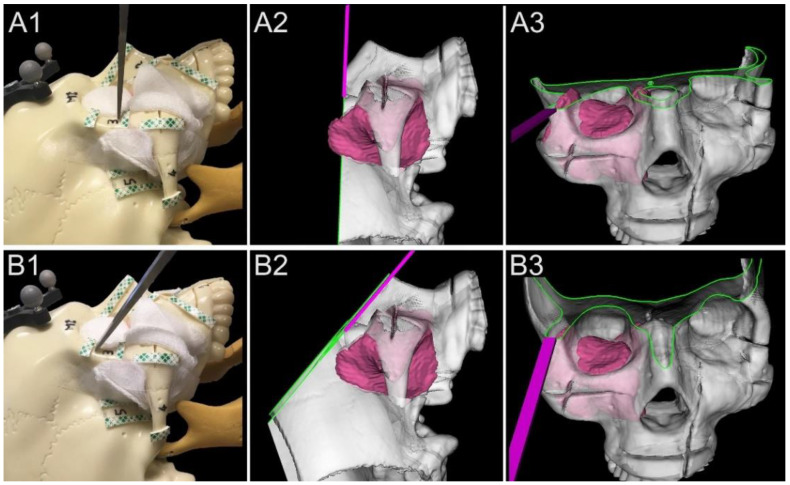
Basic principle of 3D rendering navigation for margin delineation. (**A1**–**A3**). Real lateral view, lateral 3D rendered view, and front 3D rendered view example of an unguided simulation. The virtual cutting plane crosses a portion of the tumor model located into the temporal fossa. (**B1**–**B3**). Real lateral view, lateral 3D rendered view, and front 3D rendered view of navigated simulation. With real-time 3D rendering navigation, the surgeon shifted the osteotome cranially and tilted it parallel to the surface of the tumor (green line shows the intersection between the cutting plane and bone).

**Figure 3 jcm-12-02706-f003:**
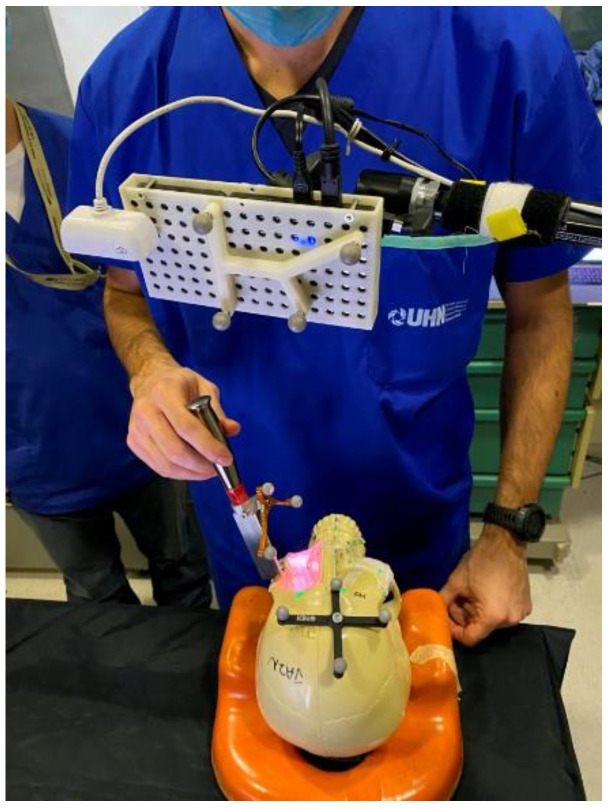
Augmented reality system mounted and projecting.

**Figure 4 jcm-12-02706-f004:**
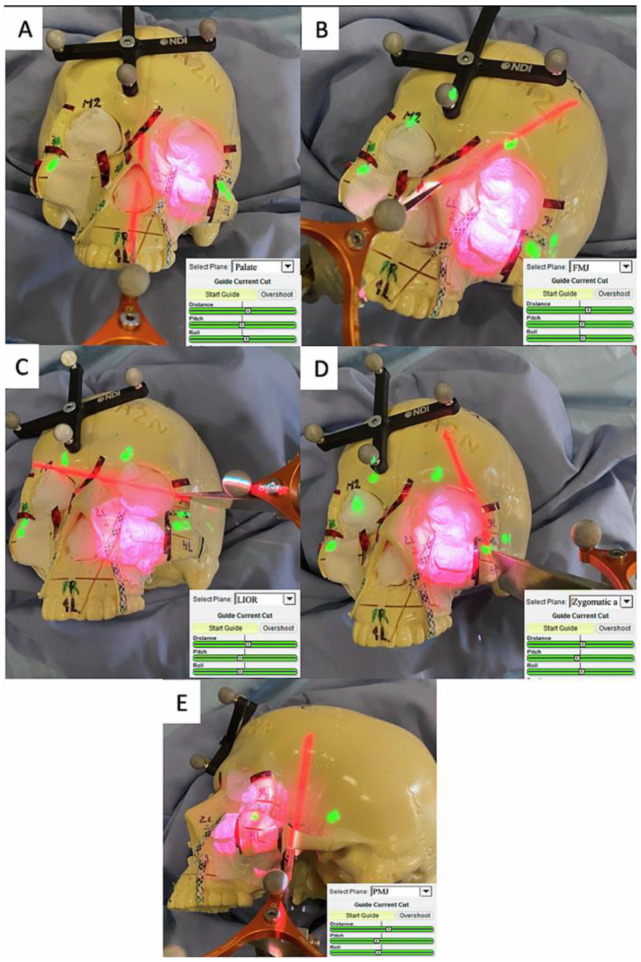
Example of the different AR projected osteotomies required to resect a left-sided maxillary tumor in sequence, with zoom-ins into the distance pitch and roll cutting parameters projected onto the surgical field: (**A**) palate, (**B**) fronto-maxillary junction, (**C**) lower inferior orbital rim, (**D**) zygomatic arch, and (**E**) pterygomaxillary junction.

**Figure 5 jcm-12-02706-f005:**
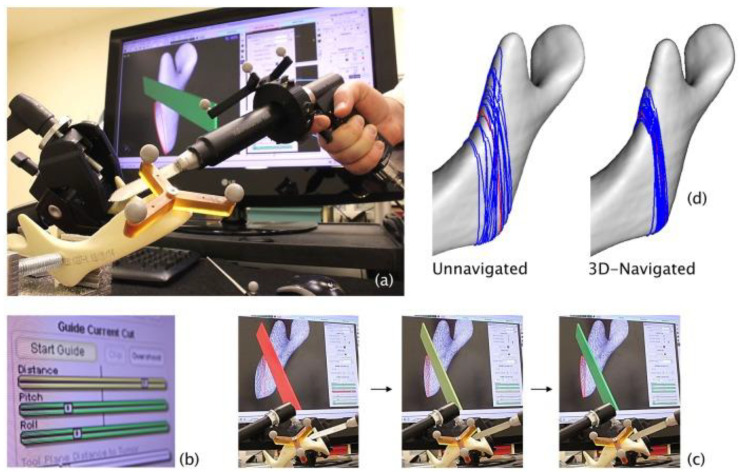
GTx-Eyes with a navigated saw (**a**) virtual cutting guide (red line) displayed on a CT reconstruction of the Sawbones mandible model, with navigated reciprocating saw correctly aligned (the saw blade is green) (bone anterior to the plane of the saw blade is clipped in the image so that the osteotomy plane can be visualized through the bone); (**b**) the indicators of distance, pitch and roll move and change in color from red to yellow and then green as the navigated saw is aligned precisely with the virtual cutting guide; (**c**) the saw blade also turns from red to yellow and then green as it is lined up correctly; (**d**) virtual cutting guide (red line) and unnavigated and 3D-navigated osteotomies (blue lines) after the analysis of multiple osteotomized models.

## Data Availability

Data sharing not applicable.

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
