# Peer review of "Skull-Base Surgery—A Narrative Review on Current Approaches and Future Developments in Surgical Navigation"

_jcm, 2023, doi:10.3390/jcm12072706_

Round 1

Reviewer 1 Report

The authors present a paper titled as a  review of current approaches and future developments in surgical navigation of skull base surgery. 

A review of the literature is limited mainly to surgical navigation in the field of head and neck surgery, however, the literature review is very cursory and does not present a thorough and comprehensive summary of published papers in this nowadays extensive field. The extent of the presented literature analysis is too scarce and cannot be accepted as a full literature review. 

Another problem is the very rudimentary presentation of their experience with an in-house navigation software package (GTxEyes) which was mostly used in pre-clinical settings, reportedly with proven benefits and in proof-of-principle clinical studies. 

 They present their  "GTx Lab experience" in the field of objective quantification of surgical volumes, development of cutting guidance system and virtual preoperative planning with augmented reality. The basic set up is shown in several illustrative and acceptable figures, however, their results were mentioned rather incidentally and there is no comprehensive discussion of comparative navigation systems, which already exist for many years in the surgical market. Even AR or MR(mixed reality) based presurgical planning systems are available nowadays, but there is no mention of other systems in this paper. 

The overall content of this paper does not actually justify the proposed title (Surgical navigation for skull base surgery: a review of current approaches and future developments)

Author Response

We thank the reviewer for taking the time to evaluate the manuscript and have revised it accordingly. 

This study was not aimed to be a formal systematic literature review but a narrative one.

However, with regards to your comment, we have first revised the manuscript title to “Skull base surgery – a narrative review on current approaches and future developments in surgical navigation”

Furthermore, we have extended our review in the surgical navigation field with a focus on skull base surgery, mainly adding clinical data.

Regarding the "GTx Lab experience", our work was done mainly in a pre-clinical setting. Early translational work from pre-clinical studies to clinical settings was done in collaboration with our orthopedic group and was added to the text. We aim to replicate these experiments within the skull base field in the near future.

Reviewer 2 Report

I read with interest the paper entitled “Surgical navigation for skull base surgery: review of current approaches and future developments”. The authors are to be commended on their work in preparation and formulating the review article. The authors work through the current state of skull base navigation and proceeds to talk about future developments and where they see the field of navigation going. The authors do not present any new data but do discuss their work in the development of a new navigation technology.

Author Response

We thank the reviewer for taking the time to evaluate the manuscript.

Reviewer 3 Report

In the manuscript entitled « Surgical navigation for skull base surgery: review of current approaches and future developments » the authors describe the concept and current applications of surgical navigation in skull base surgery, and they describe their experience in this research field.

The article is very clear, and the part on research is particularly interesting. A few suggestions :

-        Title : the authors could add « narrative » review

-        Introduction : the authors state that « Surgical navigation, together with improved surgical instrumentation and techniques, have helped in decreasing complication rates in skull base functional surgery, and extended surgical indications in oncologic procedures (11–19) ». Has the complication rate with/without SN been really assessed in skull base surgery ?

-        In part 2.2 :

o   this part is quite short, could the authors provide more data on previous studies on SN in skull base surgery ? For example details on the demonstrated advantages provided by SN (complication rate with/without SN ? benefits in terms of oncological/functional outcomes ?)

o    « In particular, surgical navigation has been shown to improve margin status in head and neck cancer. » : the authors should probably modify this sentence to something like « Preliminary studies in a limited number of patients suggest that SN may improve margin status »

-        Conclusion : « the available literature suggests that an improved intraoperative accuracy may also result in a clinical benefit in terms of outcome and reduced complication risk. » : could the authors elaborate on this in the main text ?

Author Response

We thank the reviewer for taking the time to evaluate the manuscript and have revised it accordingly. 

In the manuscript entitled « Surgical navigation for skull base surgery: review of current approaches and future developments » the authors describe the concept and current applications of surgical navigation in skull base surgery, and they describe their experience in this research field.

The article is very clear, and the part on research is particularly interesting. A few suggestions :

-        Title : the authors could add « narrative » review

Together with the first reviewer suggestions, we have revised the manuscript title to “skull base surgery - current approaches and future developments in surgical navigation”

-        Introduction : the authors state that « Surgical navigation, together with improved surgical instrumentation and techniques, have helped in decreasing complication rates in skull base functional surgery, and extended surgical indications in oncologic procedures (11–19) ». Has the complication rate with/without SN been really assessed in skull base surgery ?

      Thank you for your comment.

      Complication rates has been assessed in skull base surgery with/without surgical navigations, mainly in small sample series. The relevant studies have been added to the text.

-        In part 2.2 :

o   this part is quite short, could the authors provide more data on previous studies on SN in skull base surgery ? For example details on the demonstrated advantages provided by SN (complication rate with/without SN ? benefits in terms of oncological/functional outcomes ?)

      We appreciate the reviewer comment. We have performed a more comprehensive review to the surgical navigation section. We have added studies that elaborated on the surgical advantage of surgical navigation in the pre-clinical settings.

« In particular, surgical navigation has been shown to improve margin status in head and neck cancer. » : the authors should probably modify this sentence to something like « Preliminary studies in a limited number of patients suggest that SN may improve margin status »]

Thank you for your comment. We have changed the text accordingly.

-        Conclusion : « the available literature suggests that an improved intraoperative accuracy may also result in a clinical benefit in terms of outcome and reduced complication risk. » : could the authors elaborate on this in the main text ?

      We appreciated your comment. We have changed our manuscript according to your comments and added more data on complications and surgical outcome.

Round 2

Reviewer 1 Report

Extensive revision of the paper was done according to reviewer's suggestions